# Trend of Admissions Due to Chronic Lower Respiratory Diseases: An Ecological Study

**DOI:** 10.3390/healthcare11010065

**Published:** 2022-12-26

**Authors:** Ahmed M. Al Rajeh

**Affiliations:** Department of Respiratory Care, College of Applied Medical Sciences, King Faisal University, Al-Ahsa 31982, Saudi Arabia; amalrajeh@kfu.edu.sa

**Keywords:** admissions, chronic, diseases, England, respiratory, Wales

## Abstract

Objective: This study aimed to examine the trend of hospital admissions related to chronic lower respiratory diseases in England and Wales between 1999 and 2020. Method: This ecological analysis used data that were made accessible to the public and were taken from the Patient Episode Database for Wales (PEDW) and the Hospital Episode Statistics (HES) databases in England for the time span between April 1999 and April 2020. The patients were grouped into four age groups: under 15, 15–59, 60–74, and 75 years and above. Results: In 2020, there were 432,193 chronic lower respiratory disease hospital admissions, which increased from 239,606 in 1999. The hospital admission rate increased by 57.5% (from 459.54 (95% CI 457.71–461.38) in 1999 to 723.70 (95% CI 721.55–725.85) in 2020 per 100,000 people, *p* < 0.5). The majority of hospital admissions for chronic lower respiratory diseases were found to be directly linked to age (more prevalent in the 75+ age group). Moreover, female hospital admission rates for chronic lower respiratory diseases grew by 85.2% between 1999 and 2020, increasing from 445.45 (95% CI 442.92–447.97) to 824.96 (95% CI 821.73–828.19) per 100,000 people. Conclusion: The rate of hospital admissions due to chronic lower respiratory diseases has sharply increased during the past two decades. COPD was the most common cause for chronic lower respiratory disease admissions. Ageing was also found to be a factor in increased hospital admissions. Future studies are warranted to identify other risk factors of hospital admissions due to chronic lower respiratory diseases and specifically COPD.

## 1. Introduction

Respiratory diseases are disorders of the lungs and other respiratory system structures [1]. Infections, smoking, second-hand smoke, asbestos, radon, and other types of air pollution can all lead to respiratory diseases [1]. In the United Kingdom (UK), lung diseases affect one in five persons. Additionally, lung diseases contribute to more than 6 million inpatient bed days, 45,833 new diagnoses per month, and more than 700,000 hospital admissions per year in the UK [2]. The respiratory system can be impacted by a wide range of problems, including chronic diseases and infectious diseases. 

As described by the International Classification of Diseases 10th Revision (ICD-10), chronic lower respiratory diseases include eight chronic diseases: bronchiectasis, other chronic obstructive pulmonary diseases, simple and mucopurulent chronic bronchitis, emphysema, status asthmatics, bronchitis, not specified as acute or chronic, asthma, and unspecified chronic bronchitis [3]. The social determinants of health and health behaviours, such as obesity, cigarette smoking, exposure to occupational agents, physical inactivity, exposure to allergens, outdoor air pollution, and unhealthy diet, are the most universally recognised, significant modifiable risk factors for chronic lower respiratory diseases [4]. Chronic lower respiratory diseases are a significant cause of morbidity and mortality, ranking as the fourth leading cause of death globally [5,6]. Chronic lower respiratory diseases were the third leading cause of mortality in the UK in 2019 and were responsible for 5.9% of all deaths (31,221) [7]. The primary cause of chronic lower respiratory disease mortality and morbidity is the episodic worsening of respiratory symptoms or the exacerbations of chronic lower respiratory disorders [8]. 

The severe exacerbations that call for hospitalisation or a referral to the emergency department (ED) account for more than 50% of the costs related to chronic lower respiratory diseases [9]. Chronic lower respiratory diseases, which accounted for 26.4% of all hospital admissions for respiratory diseases in England and Wales, were the second most common reason for these admissions [10]. Understanding hospitalisation patterns for chronic lower respiratory diseases might help physicians treat patients more effectively, which reduces the number of hospital admissions associated with these diseases. Despite the intensive research that has recently been conducted in the UK by Naser and his colleagues, who examined the epidemiology of multiple acute and chronic health conditions [11,12,13,14,15], there is no previous research that examined the epidemiology of admissions related to chronic lower respiratory diseases. A recent study by Naser et al. examined the epidemiology of respiratory disease admissions in general but did not focus on hospital admissions related to chronic lower respiratory diseases [10]. This study aimed to examine the trend of hospital admissions related to chronic lower respiratory diseases in England and Wales between 1999 and 2020.

## 2. Methods

### 2.1. Study Sources and the Population

This ecological analysis used data from the Patient Episode Database for Wales (PEDW) and the Hospital Episode Statistics (HES) databases in England for the time span between April 1999 and April 2020. Hospital admission information for patients with chronic lower respiratory diseases from all age groups is available in the HES and PEDW databases. These two medical databases were previously used to examine the admission trends for different health conditions [16,17,18,19,20,21,22,23,24]. The patients in these two medical databases are grouped into four age groups: those under 15 years, those between 15 and 59 years old, those between 60 and 74 years old, and those 75 years of age and older. Using the ICD-10, 5th Edition (the classification system used by the National Health Service (NHS) to define diseases and other health problems), we were able to identify hospital admissions associated with chronic lower respiratory diseases. Chronic lower respiratory diseases (J40–J47) were used in this study.

### 2.2. Statistical Analysis

SPSS version 25 was used for all analyses (IBM Corp, Armonk, NY, USA). Using the finalised consultant episodes of hospitalisation due to chronic lower respiratory diseases divided by the mid-year population, hospital admission rates with 95% confidence intervals (CIs) were determined. To compare the variations in hospital admission rates between 1999 and 2020, we performed the chi-squared test.

## 3. Results

In 2020, there were 432,193 chronic lower respiratory disease hospital admissions, which increased from 239,606 in 1999. The hospital admission rate increased by 57.5% (from 459.54 (95% CI 457.71–461.38) in 1999 to 723.70 (95% CI 721.55–725.85) in 2020 per 100,000 people, *p* < 0.5).

Other chronic obstructive pulmonary diseases (“chronic obstructive pulmonary disease with (acute) lower respiratory infection and chronic obstructive pulmonary disease with (acute) exacerbation”) and asthma accounted for 62.3% and 26.7%, respectively, of all hospital admissions for chronic lower respiratory diseases (Table 1).

A 2.87-fold increase in the rate of hospital admissions for chronic lower respiratory diseases was observed in bronchiectasis over the duration of the study. Additionally, hospital admission rates for chronic lower respiratory diseases in the categories of other chronic obstructive pulmonary diseases, asthma, and bronchitis not classified as acute or chronic increased by 65.5%, 46.1%, and 45.2%, respectively. However, hospital admission rates for unspecified chronic bronchitis, simple and mucopurulent chronic bronchitis, status asthmaticus, and emphysema were reduced by 68.2%, 41.8%, 41.3%, and 7.8%, respectively (Figure 1, Table 2).

The age group of 75 years and above accounted for 33.8% of the total number of hospital admissions for chronic lower respiratory diseases, followed by the age group of 60 to 74 years, with 32.8%, the age group of 15 to 59 years with 24.6%, and the age group of below 15 years with 8.8%. 

Hospital admission rates due to chronic lower respiratory diseases dropped by 34.0% for patients under the age of 15 (from 330.24 (95%CI 326.67–333.82) in 1999 to 218.01 (95%CI 215.22–220.80) per 100,000 people) between 1999 and 2020. By contrast, the rate of hospital admissions among patients aged 15 to 59 years increased by 80.1% from 181.77 (95%CI 180.28–183.26) in 1999 to 327.28 (95%CI 325.38–329.19) in 2020. Moreover, in patients aged 60 to 74 years, the rate of hospital admission for chronic lower respiratory diseases increased by 34.5% (from 1136.62 (95%CI 1128.74–1144.51) in 1999 to 1528.94 (95%CI 1521.08–1536.81) in 2020 per 100,000 people). Hospital admission rates for individuals 75 years of age and older who had chronic lower respiratory diseases increased by 60.6% (from 1812.51 (95%CI 1799.29–1825.72) in 1999 to 2911.64 (95%CI 2897.15–2926.13) in 2020 per 100,000 people) (Figure 2).

During the study period, England and Wales reported a total of 6,817,120 hospital admission episodes for chronic lower respiratory diseases. Females accounted for 3,688,092 hospital admission episodes, or 54.1% of all hospital admissions related to chronic lower respiratory diseases, at a mean rate of 175,623 per year. Female hospital admission rates for chronic lower respiratory diseases grew by 85.2% between 1999 and 2020, increasing from 445.45 (95% CI 442.92–447.97) to 824.96 (95% CI 821.73–828.19) per 100,000 people. Male hospital admission rates for chronic lower respiratory diseases increased by 30.7% (from 474.31 (95% CI 471.64–476.98) in 1999 to 620.15 (95% CI 617.32–622.98) in 2020 per 100,000 people) (Figure 3).

### 3.1. Chronic Lower Respiratory Disease Admission Rate by Gender

The majority of chronic lower respiratory diseases, such as bronchitis that was not classified as acute or chronic, other chronic obstructive lung diseases, asthma, status asthmaticus, and bronchiectasis, had higher hospital admission rates in females than in males (Appendix A). However, hospital admission rates for emphysema, nonspecific chronic bronchitis, and simple and mucopurulent chronic bronchitis were greater in males than in females (Appendix A).

### 3.2. Chronic Lower Respiratory Diseases Admission Rate by Age Group

The majority of hospital admissions for chronic lower respiratory diseases were found to be directly linked to age (more prevalent in the 75 and older age group), and these diseases include bronchitis that is neither acute nor chronic, simple and mucopurulent chronic bronchitis, unspecified chronic bronchitis, emphysema, other chronic obstructive pulmonary diseases, and bronchiectasis. Even so, hospital admissions for asthma were more frequent in the following age groups: 15 years and under, 75 and older, 60 to 74 years, and 15 to 59 years, respectively. Status-asthmaticus-related hospital admissions were more prevalent in the following age groups: 60 to 74 years, 75 years and older, and those under 15 years (Appendix A).

## 4. Discussion

This was an ecological study that examined the trend of hospital admissions related to chronic lower respiratory diseases in England and Wales between 1999 and 2020. The key findings are as follows: The hospital admission rate per 100,000 people increased by 57.5% from 1999 to 2020. The majority of hospital admissions for chronic lower respiratory diseases were found to be directly linked to age (more prevalent in the 75+ age group). Female hospital admission rates increased by 85.2%, while male hospital admission rates increased by 30.7%. Other chronic obstructive pulmonary diseases (chronic obstructive pulmonary with (acute) lower respiratory infection and (acute) exacerbation), and asthma accounted for 89% of all hospital admissions for chronic lower respiratory diseases. Hospital admissions for other chronic obstructive pulmonary diseases were more frequent in the age groups of 60 to 74 years, and 75 and older, whereas asthma was more frequent in the age group of 15 years and under.

In 2019, the World Health Organization (WHO) ranked chronic obstructive pulmonary (COPD), and lower respiratory infections as the third and fourth leading causes of death globally, respectively [25]. Similarly, in the UK, chronic lower respiratory diseases ranked as the fourth leading cause of death [26]. Therefore, it is not surprising that the rate of hospital admission increased by 57.5%. However, this increase would increase the burden on the healthcare system. According to the British Lung Foundation, the UK spends GBP 11 billion a year on lung disease [27]. Around 58% of the total cost is associated with COPD (29%), lower respiratory infections (16%), and asthma (13%). This is consistent with the findings of this study, as 89% of all hospital admissions were due to COPD (62.3%) and asthma (26.7%). Despite the positive outcomes of the current practice of self-management, rescue packs, and pulmonary rehabilitation [28], the results of this study indicate that more work is needed on the management plan and that it might be possible to incorporate new health solutions, such as telehealth, which could enable early detection and quick access to medical services or interventions that could prevent hospital admission, thereby lowering the rate of hospital admissions [29,30]. 

The UK’s population in 2019 is predicted to be equal to 66.8 million, according to the Office for National Statistics release of the results of the 2021 census. The number of people 65 and older increased from 15.8% in 1999 to 18.5% in 2019 and is projected to reach 23.9% by 2039. Along with an increase in persons living alone, they also noted that rural and coastal areas have a higher percentage of senior residents. This could reduce the likelihood of having quick and simple access to healthcare support when needed or for routine care, which could cause deterioration and necessitate hospitalisation [31]. In this study, there was a linkage between the rate of admission and the age group (Appendix A). The age group of 75 years and above accounted for 33.8% of the total number of hospital admissions for chronic lower respiratory diseases, followed by the age group of 60 to 74 years with 32.8%, and the age group of 15 to 59 years with 24.6%. This could be due to the fact that ageing is associated with changes in the structure of the thoracic cavity and a reduction in the strength of the inspiratory and expiratory muscles [32,33,34]. Additionally, a decrease in the strength of the respiratory muscles could have an effect on how well the airways are cleared because a weaker cough could result in an accumulation of secretions [33]. Peak cough flow (PCF) and peak expiratory flow (PEF) were found to be higher in people who were more active, according to a study by Freitas et al. on 61 elderly healthy individuals [35]. Age-related structural changes and their effects may increase the risk of infections and hospitalisation [36]. 

In this study, the hospital admissions rate for other chronic obstructive pulmonary diseases (chronic obstructive pulmonary with (acute) lower respiratory infection and (acute) exacerbation) was higher in the age groups of 65–74 years and 75 years and above. This increase could be due to three factors, which are ageing-related FEV1 reduction, exacerbation susceptibility, and disease prognosis and severity [37]. A study conducted by Hurst et al. revealed that the severity and frequency of COPD exacerbations are related to the disease’s severity (prognosis), which may eventually raise the hospitalisation rate [38]. Evidence demonstrates that repeated or frequent exacerbations raise a patient’s level of airway inflammation during the stable state [39]. However, the quality of life is also correlated with the frequency of exacerbations [40]. The decrease in the quality of life and physical activity will impact muscle strength, which again may impact airway clearance, and the rate of hospitalisation, as cough and secretion production are associated with exacerbations [36,41,42]. Socio-economic factors could be important contributing factors that increased the rate of hospitalisation in the UK during the study period. Smoking, air pollution, allergen exposure, poor diet, obesity, and lack of physical activity are among the most important modifiable risk factors associated with chronic lower respiratory diseases [4]. In addition, previous studies in the literature reported that education level, income, occupation, and housing conditions are associated with higher rates of hospitalisation and mortality due to chronic lower respiratory diseases [43,44,45].

In this study, hospitalisation for females increased by 85.2% from 1999 to 2020. This might be due to the impact of smoking behaviours through the years or exposure to indoor biomass [28]. However, Montserrat-Capdevila et al. found that females were more susceptible than males to the harmful effects of smoking in a research study they carried out on 24,135 COPD patients in Spain. Additionally, they noted that bronchiectasis and obstructive sleep apnoea were common concomitant conditions in females [46]. Another study assessed the factors related to the frequency of exacerbation and noted that the history of asthma and the rates of previous exacerbations per year were higher in women [38]. Although comorbidities were not evaluated in this study, the rates of admissions due to bronchiectasis and asthma were also higher in females than males, which confirms the findings of previous studies. 

This study highlights the need for additional research and treatment options for lower respiratory diseases. Furthermore, the majority of the budget should be devoted to lower respiratory diseases and the elderly age group. Furthermore, it highlights the need for greater focus on other therapy modalities, such as lung rehabilitation and smoking cessation. These findings may also be used to argue for the expansion of telehealth services in order to monitor patients who have little access to medical treatment, such as those who reside in the suburbs, or to help detect any deterioration earlier. However, as comorbidities, medications, and management plans were not examined in this study, due to the ecological nature of the study design (on the population level), the results should be interpreted with caution.

## 5. Conclusions

The rates of hospital admissions due to chronic lower respiratory diseases have increased during the past two decades in England and Wales. The majority of those admissions were due to COPD. Ageing was found to be a factor in increased hospital admissions. Pulmonary rehabilitation and smoking cessation together with other validated management strategies should be reinforced. Moreover, more research should be conducted in selected groups on utilising advanced innovative solutions to enable early detection of deterioration in health.

## Figures and Tables

**Figure 1 healthcare-11-00065-f001:**
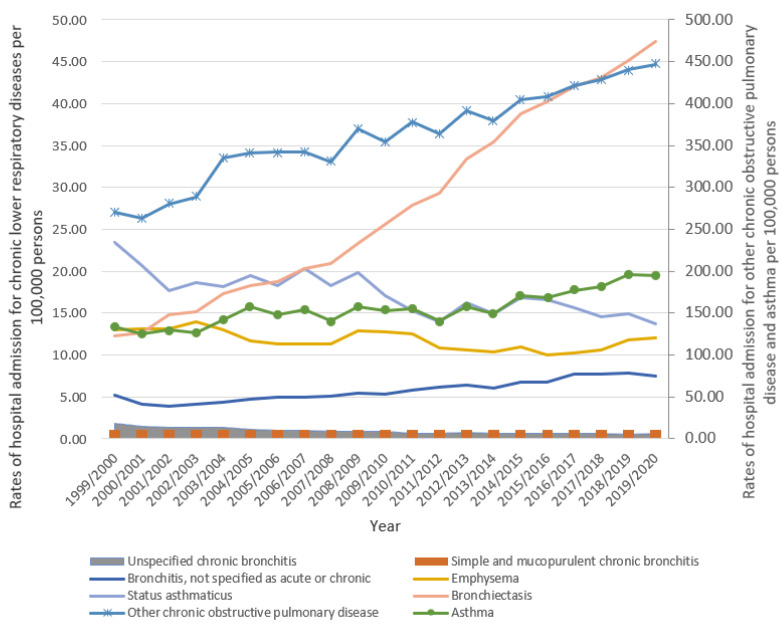
Rates of hospital admission for chronic lower respiratory diseases in England and Wales stratified by type between 1999 and 2020.

**Figure 2 healthcare-11-00065-f002:**
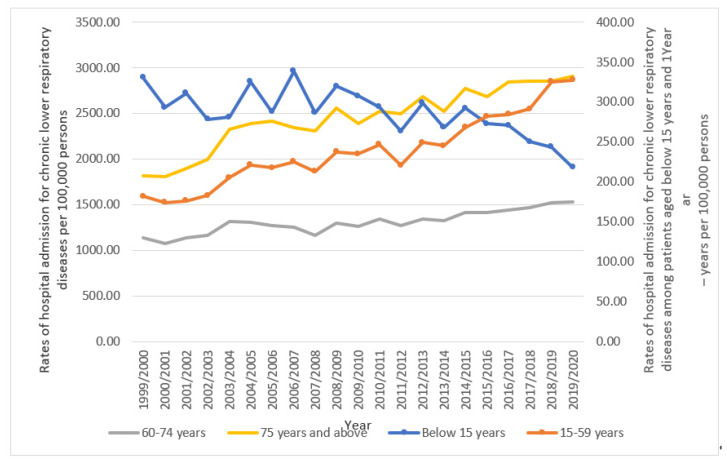
Rates of hospital admission for chronic lower respiratory diseases in England and Wales stratified by age group.

**Figure 3 healthcare-11-00065-f003:**
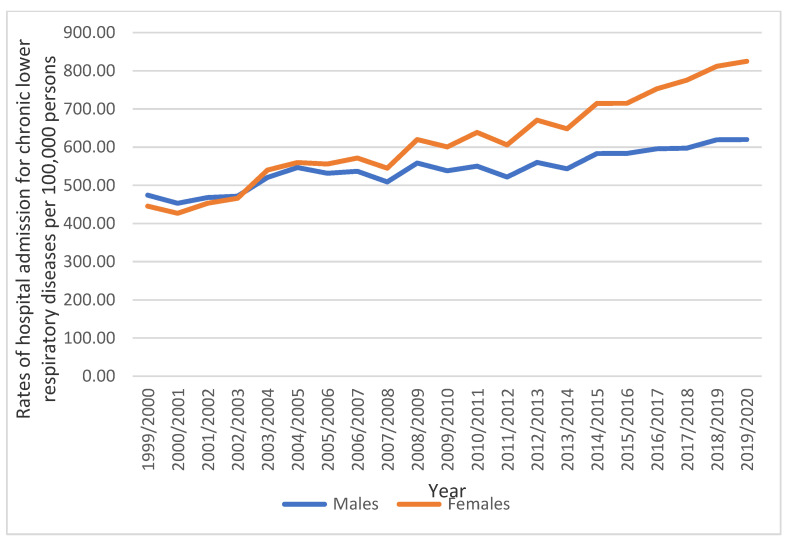
Rates of hospital admission for chronic lower respiratory diseases in England and Wales stratified by gender.

**Table 1 healthcare-11-00065-t001:** Percentage of chronic lower respiratory diseases hospital admission from total number of admissions between 1999 and 2020.

ICD Code	Description	Percentage from Total Number of Admissions
J40	“Bronchitis, not specified as acute or chronic”	1.0%
J41	“Simple and mucopurulent chronic bronchitis”	<0.1%
J42	“Unspecified chronic bronchitis”	0.2%
J43	“Emphysema”	2.0%
J44	“Other chronic obstructive pulmonary diseases (chronic obstructive pulmonary disease with (acute) lower respiratory infection and chronic obstructive pulmonary disease with (acute) exacerbation)”	62.3%
J45	“Asthma”	26.7%
J46	“Status asthmaticus”	3.0%
J47	“Bronchiectasis”	4.8%

ICD: International Statistical Classification of Diseases system.

**Table 2 healthcare-11-00065-t002:** Percentage change in the hospital admission rates for chronic lower respiratory diseases from 1999 to 2020.

Diseases	Rate of Diseases in 1999 Per 100,000 Persons (95% CI)	Rate of Diseases in 2020 Per 100,000 Persons (95% CI)	Percentage Change from 1999 to 2020	*p*-Value
“Bronchitis, not specified as acute or chronic”	5.14 (4.94–5.33)	7.46 (7.24–7.68)	45.2%	≤0.01
“Simple and mucopurulent chronic bronchitis”	0.11 (0.08–0.14)	0.06 (0.04–0.08)	−41.8%	≤0.01
“Unspecified chronic bronchitis”	1.82 (1.71–1.94)	0.58 (0.52–0.64)	−68.2%	≤0.01
“Emphysema”	12.99 (12.68–13.30)	11.98 (11.71–12.26)	−7.8%	≥0.05
“Other chronic obstructive pulmonary diseases”	270.18 (268.77–271.59)	447.18 (445.49–448.88)	65.5%	≤0.01
“Asthma”	133.59 (132.60–134.58)	195.24 (194.12–196.36)	46.1%	≤0.01
“Status asthmaticus”	23.45 (23.04–23.87)	13.77 (13.47–14.06)	−41.3%	≤0.01
“Bronchiectasis”	12.26 (11.96–12.56)	47.43 (46.88–47.98)	287.0%	≤0.001

## Data Availability

Publicly available datasets were analysed in this study. These data can be found at http://http//content.digital.nhs.uk/hes and https://www.wales.nhs.uk/document/176173#:~:text=The%20Patient%20Episode%20Database%20for%20Wales%20%28PEDW%29%20is,major%20operations%2C%20and%20hospital%20stays%20for%20giving%20birth (accessed on 14 September 2022).

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
