# Peer review of "Trend of Admissions Due to Chronic Lower Respiratory Diseases: An Ecological Study"

_healthcare, 2022, doi:10.3390/healthcare11010065_

Round 1
Reviewer 1 Report
Ahmed M Al Rajeh aimed to examine the trend of hospital admissions related to chronic lower respiratory diseases in England and Wales between 1999 and 2020. The study is interesting and may point out the unmet need for adequate COPD treatment. However, there are a few points that have to be addressed in order for the article to be accepted for publication in this journal.
Major comments
1. The author should better discuss all the possibilities of an increase in hospitalization during the investigated periods. For example, could it possibly be due to socio-economic factors, aging of the general world population (higher incidence of individuals over 75 years of age), better health care, or availability of health care?
2. It would be useful if the author described better the J44 ICD code, and the majority of patients belong to this category. Why is it named other chronic pulmonary diseases? Is that related to COPD? That may be useful for scientists without a clinical background.
Minor comments
1. If the journal formatting guidelines allow, I would suggest the author remove the subtitles from the abstract and format it so that it flows better. I believe that it would make the abstract easier to read and more appealing to the reader.
2. Why did the author focus on hospital admissions in England and Wales and not in some other geographic region? Could the author compare these data with some other regions or worldwide trends?
Author Response
Major comments
- The author should better discuss all the possibilities of an increase in hospitalization during the investigated periods. For example, could it possibly be due to socio-economic factors, aging of the general world population (higher incidence of individuals over 75 years of age), better health care, or availability of health care?
- Thank you for this comment, I totally agree with the reviewer that all the above mentioned factors could have contributed to the increase in the hospitalisation rate in this study. In the discussion section in page 7 I have discussed the role of ageing, and availability of better healthcare on hospitalization, see lines 185-216. I have now discussed the role of socio-economic factors in the discussion section.
- It would be useful if the author described better the J44 ICD code, and the majority of patients belong to this category. Why is it named other chronic pulmonary diseases? Is that related to COPD? That may be useful for scientists without a clinical background.
- Thank you for this comment, the full description for the J44 ICD codes is mentioned in Table 1, which includes Chronic obstructive pulmonary disease with (acute) lower respiratory infection and chronic obstructive pulmonary disease with (acute) exacerbation.
Minor comments
- If the journal formatting guidelines allow, I would suggest the author remove the subtitles from the abstract and format it so that it flows better. I believe that it would make the abstract easier to read and more appealing to the reader.
- Thank you for this comment, as per the journal formatting guidelines, the subtitles are necessary upon presenting the abstract.
- Why did the author focus on hospital admissions in England and Wales and not in some other geographic region? Could the author compare these data with some other regions or worldwide trends?
- Thank you for this comment, I focused on hospital admissions in England and Wales due to the availability of the data for these geographic locations. Unfortunately, I don’t have data for other regions.
Reviewer 2 Report
Abstract: The purpose and results of the study are clearly stated. I would prefer a more concise exposition, perhaps indicating only the most significant figures.
Introduction: The purpose of the study is again clearly defined. I appreciated the description of the current situation in the UK and which respiratory diseases are referred to in this discussion.
Methods: Methods are clearly stated. I simply ask to explain if there is a logical reason regarding the ages used to subdivide the patients or if they are arbitrary values.
Results: Outcomes are comprehensively defined. The presence of tables and graphs helps understanding the text.
Discussion: hypotheses are formulated regarding the variation of the numbers collected between 1999 and 2020. I appreciated the presence of suggestions that could allow us to deepen and improve the work carried out for the realization of this study.
Conclusion: the conclusions are clear and concise, taking up the fundamental points of the article.
Author Response
Abstract: The purpose and results of the study are clearly stated. I would prefer a more concise exposition, perhaps indicating only the most significant figures.
- Thank you for this comment, in the abstract I presented the main study findings to make it more interesting for the readers.
Introduction: The purpose of the study is again clearly defined. I appreciated the description of the current situation in the UK and which respiratory diseases are referred to in this discussion.
Methods: Methods are clearly stated. I simply ask to explain if there is a logical reason regarding the ages used to subdivide the patients or if they are arbitrary values.
- Thank you for this comment, the reason regarding the age categories used is that the two medical databases report and present their data stratified using these four age groups only.
Results: Outcomes are comprehensively defined. The presence of tables and graphs helps understanding the text.
Discussion: hypotheses are formulated regarding the variation of the numbers collected between 1999 and 2020. I appreciated the presence of suggestions that could allow us to deepen and improve the work carried out for the realization of this study.
Conclusion: the conclusions are clear and concise, taking up the fundamental points of the article.